# Antiviral Properties of Silver Nanoparticles against SARS-CoV-2: Effects of Surface Coating and Particle Size

**DOI:** 10.3390/nano12060990

**Published:** 2022-03-17

**Authors:** Qinghao He, Jing Lu, Nian Liu, Wenqing Lu, Yu Li, Chao Shang, Xiao Li, Ligang Hu, Guibin Jiang

**Affiliations:** 1State Key Laboratory of Environmental Chemistry and Ecotoxicology, Research Center for Eco-Environmental Sciences, Chinese Academy of Sciences, Beijing 100085, China; qhe001@outlook.com (Q.H.); yuli_st@rcees.ac.cn (Y.L.); gbjiang@rcees.ac.cn (G.J.); 2Changchun Veterinary Research Institute, Chinese Academy of Agricultural Sciences, Changchun 130122, China; lujing0819@126.com; 3School of Environment, Hangzhou Institute for Advanced Study, University of Chinese Academy of Sciences, Hangzhou 310024, China; liunian_goodluck@126.com; 4School of Life Sciences, Hebei University, Baoding 071002, China; moonlewini@outlook.com; 5School of Environment and Health, Jianghan University, Wuhan 430056, China

**Keywords:** AgNPs, SARS-CoV-2, cell activity, viral replication

## Abstract

Coronavirus disease 2019 (COVID-19) has spread rapidly and led to over 5 million deaths to date globally. Due to the successively emerging mutant strains, therapeutics and prevention against the causative virus, severe acute respiratory syndrome coronavirus 2 (SARS-CoV-2), are urgently needed. Prevention of SARS-CoV-2 infection in public and hospital areas is essential to reduce the frequency of infections. Silver nanoparticles (AgNPs) with virucidal effects have been reported. Therefore, we investigated the virucidal activity and safety of ten types of AgNPs with different surface modifications and particle sizes, in cells exposed to SARS-CoV-2 in vitro. The AgNPs could effectively inhibit the activity of SARS-CoV-2, and different surface modifications and particle sizes conferred different virucidal effects, of which 50-nm BPEI showed the strongest antiviral effect. We concluded that the efficacy of each type of AgNP type was positively correlated with the corresponding potential difference (R^2^ = 0.82). These in vitro experimental data provide scientific support for the development of therapeutics against COVID-19, as well as a research basis for the development of broad-spectrum virucides. Given the increasing acquired resistance of pathogens against conventional chemical and antibody-based drugs, AgNPs may well be a possible solution for cutting off the route of transmission, either as an external material or a potential medicine.

## 1. Introduction

Since December 2019, the acute respiratory infectious disease caused by severe acute respiratory syndrome coronavirus 2 (SARS-CoV-2) has progressed into the pandemic of coronavirus disease 2019 (COVID-19), which severely threatens public health and safety worldwide [1]. SARS-CoV-2 is spherical, with a diameter of about 100 nm. It is a single-stranded positive-strand RNA virus with an envelope [2,3,4,5], and belongs to the family of beta coronaviruses. Infection with this virus can lead to a highly contagious, acute pulmonary disease. By the end of 2021 [6,7,8], the number of the global cases exceeded 422 million, with the death toll over 5.8 million. These figures are still showing sharply rising trends in most countries [9]. To date, the US Food and Drug Administration (FDA) has fully approved only one virucide, Veklury (Remdesivir), which can only be used in healthcare centers that can provide acute in-patient care. The other drugs that may be administered are done so under the emergency use authorization of the FDA [10]. Meanwhile, the ongoing evolution of the RNA virus, exponentially, results in many SARS-CoV-2 variants, with structural changes that affect the viral properties, such as the transmission rate, associated disease severity, and immune escape [11]. With the predominance of the variants changing, medicine development has become more complicated, and, thus, the COVID-19 pandemic has yet not been effectively taken under control. In addition, contaminated fomites and airborne particles are both responsible for the high transmission rate [12]. SARS-CoV-2 remains viable and infectious in the environment for days, and has strong stability on plastic and stainless steel [13]. Therefore, effective disinfection of the environment from SARS-CoV-2 is expected to address COVID-19, and the development of new virucides with a wide target spectrum will significantly contribute to this end.

In recent years, nanoparticles, including nano-scale metal materials, such as those made of silver, have become the focus of virucide research, due to their unique physical and chemical properties. Currently, AgNPs have been applied as therapeutics in antimicrobial ointments, long-term burn care products, and antibacterial lotions, as well as in medical dressings (commercial products approved by the FDA), including PolyMem Silver™ (Aspen), Acticoat™, Bactigras™ (Smith & Nephew), Tegaderm™ (3M), and Aquacel™ (ConvaTec) [14,15,16,17]. Silver nanoparticles (AgNPs) are silver clusters with a size of 1–100 nm, formed through nano-crystallization of elementary silver using nano-technology. The small size (1–100 nm) of AgNPs allows them to interact with a similarly sized biological target, whereby they can pass through the typical plasma membrane barrier [18], showing excellent performance in many aspects [19].Their clinical use, especially their virucidal activity, has been evaluated in several studies. The medical properties of silver were recognized very early in the history of civilization [16]. In Europe during the middle ages, compounds containing silver were explicitly recorded as drugs in the Roman Pharmacopoeia [20]. Since the Tang Dynasty in China, elementary silver has been used to treat acne diseases caused by viruses [21,22]. In early China, the usage of elementary silver as a medicine was widely recognized. For example, Li Shizhen of the Ming Dynasty emphasized in the ‘Compendium of Materia Medica’ that only delicate foils of silver, and not any other form, should be used as a medicine in case of toxicity [23]. Such ancient records underscore the potential of nano-silver materials in the field of virucides.

AgNPs are widely applied in contemporary medicine as well. Their bactericidal effect has prompted researchers in the field of virucides to assess whether these particles also have a virucidal effect [24]. AgNPs can effectively inhibit RNA and DNA viruses in vitro, such as human immunodeficiency virus, HIV-1, hepatitis B, Tacaraibe, adenovirus, parainfluenza virus, and influenza virus (H3N2) [25,26,27,28,29]. Studies have shown that AgNPs mainly exert their virucidal effect through two mechanisms. The first mechanism aims to prevent the virus from infecting the host cell, and this effect can be achieved through a combination of sulfur-containing residues on the surface glycoprotein [30]. For example, Morris et al. found that, via attaching to the viral glycoprotein, AgNPs can prevent RSV from entering the host cells [31]. In the second mechanism, AgNPs enter the host cell and then block the machinery necessary for viral assembly [32,33]. Lv et al. found that AgNPs reduce TEGV-induced apoptosis, by modulating the p38/mitochondrial caspase3 signaling pathway [34]. Compared with plain AgNPs, coated AgNPs are reported to increase stability, and decrease cytotoxicity and agglomeration [19]. For instance, a polyvinylpyrrolidone (PVP) coating facilitates the usage of AgNPs in antiviral applications and lowers the associated cytotoxicity [35]. 

After the spread of the COVID-19 pandemic for three years, a promising drug for combating it is still lacking. The rapid mutation of SARS-CoV-2 leads to a substantial diminishing of the neutralizing activity of many existing COVID-19 vaccines and therapies. There is an urgent need to develop drugs against SARS-CoV-2 via new mechanisms. Metal-based antimicrobial agents have been used for a long time and possess antiviral activity towards various viruses. The nanoform of these metals have demonstrated even better activities in recent years. Several studies have shown that AgNPs can play preventive role against SARS-CoV-2 infection and can contribute to the development of vaccines and virucides based on nanotechnology [36]. For example, in locations with severe viral or bacterial infections, such as healthcare facilities, nano-silver–modified titanium dioxide is used for self-disinfection of the environment through photocatalysis, thereby preventing the spread of SARS-CoV-2 [37]. Jeremiah et al. evaluated AgNPs with different sizes and concentrations and found that 1–10 ppm of the particles with a diameter of approximately 10 nm could effectively neutralize extracellular SARS-CoV-2, and cytotoxicity was observed at >20 ppm [19]. Al-Sanea et al. applied strawberry and ginger extracts in the fabrication of AgNPs, to enable the anti-SARS-CoV-2 activity [38]. Surface modification of AgNPs can significantly affect their biocompatibility and their interaction with cells [39]. House et al. observed that AgNPs modified with branched BPEI had a more significant gene-regulatory effect than unmodified particles [40].

In this study, AgNPs with different surface modifications were assessed for their virucidal effects against SARS-CoV-2 in vitro, and three types of surface coatings that differed widely in their surface charge were selected as the modification materials, namely citrate, polyvinyl pyrrolidone (PVP), and branched polyethylenemine (BPEI). A series of analytical methods were adopted, such as the CCK-8 method and flow cytometry, to assess cell viability and apoptosis, respectively. In addition, crystal violet staining, Hoechst staining, virus titration, and reverse transcription–quantitative polymerase chain reaction were carried out to assess the safety and anti–SARS-CoV-2 effect of each type of AgNP in vitro. Our results indicate that these nanoparticles can serve as potential virucides against SARS-CoV-2 infection.

## 2. Methods and Materials

### 2.1. Chemicals, Viral Strain, and Cell Lines

The SARS-CoV-2 strain BetaCoV/Beijing/IME-BJ01/2020 (Beijing, China) was originally isolated, as previously described [41]. Virus stocks were propagated and tittered on African green monkey kidney (Vero E6) cells. All experiments with infectious SARS-CoV-2 were conducted in a laboratory at biosafety level 3 in Changchun Veterinary Research Institute. Vero E6 cells were preserved in our laboratory. The cells were cultured at 37 °C with 5% CO_2_ in Dulbecco’s modified Eagle medium (DMEM) supplemented with 10% fetal bovine serum (FBS), 50 U mL^−1^ of penicillin, and 50 µg mL^−1^ streptomycin. Ten types of AgNPs, including PVP-coated AgNPs with diameters of 5, 20, 50, and 100 nm, citrate-coated AgNPs with diameters of 5, 20, 50, and 100 nm, and BPEI-coated AgNPs with diameters of 50 and 100 nm, were purchased from nanoComposix (San Diego, CA, USA). 

### 2.2. Characterization of the AgNPs

The size distribution and zeta potential of each type of AgNP were characterized. The AgNPs were sonicated for approximately 5 min before each characterization. The hydrodynamic size distributions and surface zeta potentials of the AgNPs in Milli-Q water were measured via dynamic light scattering using a Malvern Zetasizer Nano ZS (Malvern Panalytical, Malvern, UK). The hydrodynamic size distribution and zeta potential of the AgNPs were measured at least three times. High-resolution transmission electron microscopy (HRTEM, JEOL Ltd., Tokyo, 2100F, Japan) with an accelerating voltage of 200 keV was used to characterize the physical size distribution of the AgNPs with a copper mesh. The AgNPs solution (8 µL) was dropped onto a 200-mesh ultra-thin carbon-coated copper grid (Beijing Zhongjingkeyi Technology Co., Ltd., Beijing, China). More than 200 particles were examined per sample, and the particle diameters were measured using Image J software (Image J 1.53e, National Institutes of Health, Bethesda, MD, USA). 

### 2.3. Cell-Viability Assay

Cell viability was assessed using a Cell Counting Kit-8 (CCK-8; CK04, Dojindo Laboratories, Kumamoto, Japan). Vero E6 cells (5 × 10^3^) were seeded into each well of a 96-well plate and cultured for 24 h before any treatment. The drug treatment group was administered various concentrations of each modified particle at various sizes. After 2 h of incubation, the cells were infected with SARS-CoV-2 for 48 h. Subsequently, 10 μL CCK-8 reagent was added, and after 2 h of incubation at 37 °C, the optical density of the samples at 450 nm wavelength was measured using a microplate reader.

### 2.4. Crystal-Violet Staining

Cell proliferation was evaluated using crystal violet staining. Vero E6 cells (2 × 10^5^) were seeded in each well of 6-well plates and cultured for 24 h. After 48 h of drug treatment, the medium of the culture was replaced with crystal violet dye, and the samples were further incubated for 15 min. Subsequently, the dye was removed, and the cells were washed with phosphate-buffered saline (PBS) and then photographed.

### 2.5. Hoechst Staining

Vero E6 cells (2.5 × 10^5^) were seeded in each well of 6-well plates and cultured for 24 h. After 48 h of drug treatment, the medium was removed, the cells were washed three times with PBS, and 1 mL of 1:1000 diluted Hoechst dye was added to each well. After incubating the samples for 15 min in the dark, they were rinsed twice with PBS, and then their nuclear morphology was observed using a fluorescence microscope, and the samples were photographed.

### 2.6. Annexin-V Analysis

Vero E6 cells (5 × 10^5^) were plated into each well of 6-well plates and cultured for 24 h. Subsequently, they were treated with each type of AgNP for 48 h. Afterward, the cells were collected via centrifugation at 1500 rpm/min for 5 min, resuspended in 500 μL 1× Annexin V Binding Buffer, and then 5 μL FITC and 5 μL propidium iodide were added. The samples were incubated for 15–20 min in the dark and then assessed for their apoptosis via flow cytometric analysis.

### 2.7. Reverse Transcription–Quantitative Polymerase Chain Reaction (RT-qPCR)

The viral RNA in culture supernatants was extracted using a QIAamp Viral RNA Kit (Qiagen, Hilden, Germany). Virus copies were then detected using RT-qPCR with a HiScript II One Step qRT-PCR SYBR Green Kit (Vazyme Biotech, Nanjing, China). The protocol for the RT-qPCR was as follows: 50 °C for 15 min, 95 °C for 30 s, followed by 45 cycles at 95 °C for 10 s and 63 °C for 35 s. The primers used to detect the SARS-CoV-2N gene were as follows:

N gene Forward: 5′-GGGGAACTTCTCCTGCTAGAAT-3′;

Reverse: 5′-CAGACATTTTGCTCTCAAGCT-3′;

E gene Forward: 5′-CGATCTCTTGTAGATCTGTTCTC-3′.

Reverse: 5′-ATATTGCATTGCAGCAGTACGCACA-3′.

The sequences of the TaqMan probes were as follows: 

N gene 5′-FAM-TTGCTGCTGCTTGACAGATT-TAMRA-3′

E gene 5′-ATATTGCATTGCAGCAGTACGCACA-3′.

### 2.8. Measurement of Viral Titers

Vero E6 cells (5 × 10^3^) were seeded in each well of 96-well plates and cultured for 24 h. The culture supernatant was continuously diluted 10 times in serum-free DMEM with a volume of 100 μL/well, and then 100 μL DMEM with 4% FBS, 100 IU/mL penicillin, and 100 μg/mL streptomycin were added 2 h post-infection, to reach a final volume of 200 μL per well. The culture plate was cultured at 37 °C with 5% CO_2_. The number of cytopathic pores was observed and recorded every day. The virus titer was calculated using the Reed–Muench method.

### 2.9. Statistical Analyses

All data were analyzed using GraphPadPrism version 8.0 (GraphPad Software Inc., San Diego, CA, USA) and expressed as mean ± SEM. The significance of each difference was assessed using the *t*-test or ANOVA followed by the two-tailed *t*-test. A *p*-value of <0.05 was considered to indicate statistical significance.

## 3. Results

### 3.1. Characteristics of the AgNPs

The morphology of AgNPs was nearly spherical in TEM images (Figure 1A–J). The size distributions of AgNPs in TEM images displayed narrow normal distributions (Appendix A), which showed that the sizes of the AgNPs were close to the values advertised by the manufacturer. The hydrodynamic size distribution displayed an almost unimodal distribution (Appendix A), indicating uniform dispersion of AgNPs. Zeta potential reflects the surface charge of AgNPs and differed with the particle size and surface coating. The zeta potentials of the PVP-coated and citrate-coated AgNPs showed negative charges, ranging from –10.8 ± –27.4 mV to –3.1 ± –31.5 mV, whereas the BPEI-coated AgNPs showed a positive charge (Appendix A). The chemical structure of each AgNP surface coating is shown in Appendix A. The detailed characteristics of the AgNPs are listed in Table 1.

### 3.2. Effects of AgNPs on the Viability of Vero E6 Cells Infected with SARS-CoV-2

After ascertaining the characteristics of the AgNPs, we further carried out experiments on infected cells. By detecting cell proliferation through CCK-8 experiments, we aimed to determine the cell protective effect and biological friendliness of AgNPs, and study the effects of particle size and surface modification on the cell protective effect. The viability of Vero E6 cells significantly decreased after SARS-CoV-2 (MOI = 0.008) infection, as shown in Figure 2. However, the three types of AgNPs significantly improved the viability of the infected cells. The viability of Vero E6 cells infected with SARS-CoV-2 with AgNPs at various concentration was determined (Appendix A). Compared with the SARS-CoV-2 treatment group, at a concentration of 1 μg/mL, the viability of cells was increased by 7.18%, 15.37%, 7.93%, and 8.96% in the 5, 20, 50, and 100 nm citrate groups, respectively (Figure 2A); increased by 16.01%, 15.48/%, 10.67%, and 12.32% in the 5, 20, 50, and 100 nm PVP groups, respectively (Figure 2B); and increased by 20.70% and 16.72% in the 50 and 100 nm BPEI groups, respectively (Figure 2C). Furthermore, smaller AgNPs had more significant effects in improving the viability of infected cells than large AgNPs. Additionally, 50-nm citrate-modified and PVP-modified AgNPs increased the cell viability by 11.49% and 9.72%, respectively, compared with the level in the control group. The AgNPs with BPEI modification increased cell viability by 20.70% (Figure 2D). AgNPs is an ultramicroscopic material, typically ranging in size from 1 to 100 nm. The size and surface modification of AgNPs play key roles in the antiviral activity, and previous studies have shown that the smaller the size of the nanomaterial, the more surface reactivity is produced [42]. Jeremiah et al. evaluated a large number of different sizes and concentrations of AgNPs and observed that AgNP with a diameter of 10 nm was effective in inhibiting the activity of SARS-CoV-2 at concentrations ranging from 1 ppm to 10 ppm [19]. Our results showed the AgNPs could effectively inhibit the activity of SARS-CoV-2, and different surface modifications and particle sizes conferred different virucidal effects.

### 3.3. Effects of AgNPs on the Replication of SARS-CoV-2 in Vero E6 Cells

AgNP material has a protective effect on infected cells. In order to further verify the direct inhibitory effect of AgNPs on SARS-CoV-2 and the correlation with particle size and surface modification, we detected virus replication through virus titration experiments and RT-qPCR. The viral titer was significantly reduced in SARS-CoV-2–infected Vero E6 cells that were treated for 24 or 48 h with various sizes of citrate AgNPs or those modified with PVP or BPEI, compared with the level in infected cells not treated with any AgNP (Figure 3A–C). The number of copies of the N and E genes of SARS-CoV-2 in the AgNP-treated infected cells was significantly decreased compared with the control levels (Figure 3D–I), indicating that AgNPs significantly inhibited the replication of SARS-CoV-2 in Vero E6 cells. Furthermore, it was found that the virus titer decreased further with the decrease in the size of the AgNPs. The AgNPs modified with BPEI and PVP showed stronger virucidal effects than citrate AgNPs (Appendix A). Interestingly, the zeta potential of the modification material was positively correlated with the virus titer (R^2^ = 0.8162, *p* < 0.001), as shown in Figure 4.

### 3.4. Evaluation of the Cytotoxicity of AgNPs

AgNPs can significantly inhibit virus replication and improve the cell proliferation of SARS-CoV-2-infected cells in vitro. In order to further study the cytotoxicity of AgNPs, we carried out experiments on uninfected cells. We first analyzed the effect of AgNO_3_ on Vero E6 cell viability using the CCK-8 detection method. Compared with the control level, the viability of the Vero E6 cells was significantly reduced after AgNO_3_ (10 μg/mL) treatment (Figure 5A). To assess the toxicity of the AgNPs, Vero E6 cells treated with AgNPs of various sizes were analyzed via crystal violet staining and Hoechst staining. The results demonstrated that, although AgNO_3_ was substantially cytotoxic, no obvious cytotoxicity was detected in cells treated with any of the AgNPs (Figure 5B,C). Finally, the effects of the AgNPs on the apoptosis of Vero E6 cells were assessed using flow cytometric analysis. Compared with the apoptosis rate in the control group, the AgNP treatments did not have any significant effect (Figure 5D). These results suggested that the AgNPs have good biosafety. 

## 4. Discussion

In this study, the virucidal effects of AgNPs on SARS-CoV-2 were analyzed at non-toxic concentrations. It was discovered that the number of copies of the N and E genes and the virus titer in SARS-CoV-2–infected Vero E6 cells treated with any of the AgNPs were significantly reduced, while the cell viability was significantly improved, compared with the levels in the infected cells not treated with any AgNPs.

AgNPs can contain various surface modifications. We assessed the anti–SARS-CoV-2 virucidal effects of three differently modified types of AgNPs. The AgNPs coated with PVP or BPEI had stronger virucidal effects than those coated with citrate. More importantly, the zeta potential of the modification material was positively correlated with the virucidal effect, providing a new insight for further research on the virucidal mechanism of AgNPs in the future. 

AgNPs are typical metal nanomaterials, with large specific surface areas (inversely proportional to the particle diameter). The smaller the particle size, the higher the percentage of surface atoms, and the more unsaturated bonds. This structure provides many adsorption and reaction sites for various types of reactions, and these particles can easily be combined with foreign atoms, through chemical bonding. Thus, their ability to attach to viruses and undergo a chemical reaction with them is greatly improved. Previous studies have proven that the virucidal effects of AgNPs depend on the size of the nanoparticles. It has been reported that AgNPs with a diameter of 10 nm have the greatest virucidal effect [30], consistent with our results. We selected three types of AgNPs with different particle sizes (5–100 nm) to evaluate their virucidal effects. According to the results of the CCK-8 and RT-qPCR assays and the viral titer estimates, the small nanoparticles had more significant virucidal effects than the large nanoparticles. Nanoparticles with different shapes have different surface-to-volume ratios, and their protein corona could be distinct. This could influence the cellular uptake of nanoparticles and their interactions with the biological system. In short, AgNPs can effectively inhibit SARS-CoV-2 infection, and AgNPs with different surface modifications or particle sizes have different virucidal effects, providing a research basis for the future development of an anti–SARS-CoV-2 virucidal.

Moreover, it is worth noting that we aimed to investigate the usage of AgNPs as a potential internal material. Accordingly, the experiments on the virus-infected cell were stricter than that on virus alone, to probe both the antiviral effects and the biotoxicity of AgNPs. We set higher standards and chose virus-infected cells to evaluate the antiviral effects of AgNPs. Fortunately, the data from our experiments presented a positive result, that the AgNPs were candidate drugs, with both biological safety and antiviral ability.

The application potential of silver-based biocomposites in antibacterial coatings, food preservation, air purification, medical devices, clinical protective clothing, and other fields has been extensively studied [43]. Given the rapidly spreading and rampant COVID-19 pandemic, the development of new virucides is critical, and low-toxicity biocompatible AgNPs may serve this end [44]. For example, AgNPs incorporated into polycotton fabrics have been found to be effective in inhibiting SARS-CoV-2 [45]. As an external antiviral agent, there will be the risk of contact with human skin and the mucous membrane and inhalation into the respiratory tract. In this study, experiments were carried out on infected cells to determine the antiviral effect and biological friendliness of AgNPs. On the other hand, AgNP has the potential for internal use. Previous studies have shown that AgNPs effectively reduced the replication of respiratory syncytial virus and the production of pro-inflammatory cytokines in epithelial cell lines and mouse lung [31]. The present study also found a direct SARS-CoV-2 inhibition and cytoprotective effect of AgNPs in infected cells, which provided a preliminary experimental basis for the use of AgNPs as internal anti-SARS-CoV-2 drugs. A previous study showed that the potential use of nanoparticles as novel antiviral therapeutics presented a lower likelihood of acquired drug resistance than that of chemical- or antibody-based antiviral therapeutics [17]. Moreover, our results showed the safety of certain AgNPs and provide supporting evidence for their virucidal application on fomites. However, for further therapeutic application, additional toxicity studies should be performed [13]. In particular, the chronic toxicity of AgNPs should be assessed, considering their close connection with human beings.

## 5. Conclusions

This study primarily aimed to evaluate the virucidal efficacy and safety of AgNPs in the setting of SARS-CoV-2 infection in vitro. Researchers evaluated the inhibitory effects of three different AgNPs (citrate, PVP, and BPEI) in host cells infected with SARS-CoV-2 in vitro. The results suggest that, although AgNO_3_ alone is toxic to Vero E6 cells, AgNPs show significantly reduced toxicity. Furthermore, the results indicate that AgNPs do not induce apoptosis. These results suggest that the three types of AgNPs are relatively safe, and the virucidal capacity is positively correlated with the potentials of the modification materials. AgNPs are less toxic to cells than silver ions, which is more conducive to the research of silver in the field of virucides. AgNPs have stronger anti–SARS-CoV-2 activity. Our results provide insights into the virucidal application of AgNPs to prevent transmission, such as their use in mouthwashes and nasal rinse solutions, as well as in personal protective equipment [46] and into their therapeutic application to treat infected individuals. Therefore, these nanomaterials warrant further studying as anti-SARS-CoV-2 agents.

## Figures and Tables

**Figure 1 nanomaterials-12-00990-f001:**
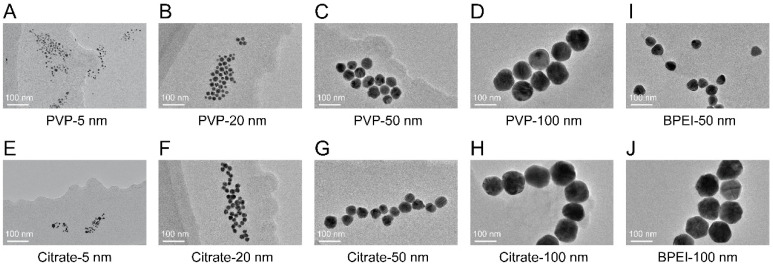
HRTEM image of AgNPs. (**A**–**D**) The morphology of PVP-coated AgNPs with different particle sizes: 5 nm (**A**), 20 nm (**B**), 50 nm (**C**), and 100 nm (**D**); (**E**–**H**) the morphology of citrate-coated AgNPs with different particle sizes: 5 nm (**E**), 20 nm (**F**), 50 nm (**G**), and 100 nm (**H**); (**I**,**J**) the morphology of BPEI-coated AgNPs with different particle sizes: 50 nm (**I**) and 100 nm (**J**).

**Figure 2 nanomaterials-12-00990-f002:**
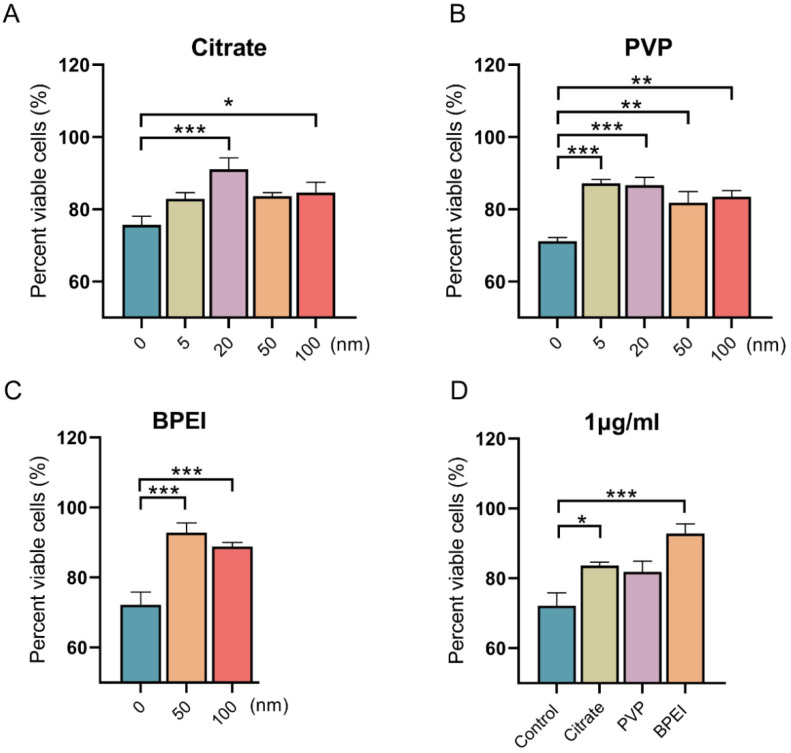
Effects of AgNPs on the viability of Vero E6 cells infected with SARS-CoV-2. Vero E6 cells were infected with SARS-CoV-2 (MOI = 0.008); 1 μg/mL AgNPs-coated with citrate, PVP, or BPEI with different particle sizes were administered for 48 h. (**A**–**C**) Effects of different particle size materials on the viability of Vero E6 cells infected with SARS-CoV-2, Citrate (**A**), PVP (**B**), BPFI (**C**); (**D**) the viability of Vero E6 cells infected with SARS-CoV-2, treated with 1 μg/mL 50 nm AgNPs; (n = 8 groups), * *p* < 0.05, ** *p* < 0.01, *** *p* < 0.001.

**Figure 3 nanomaterials-12-00990-f003:**
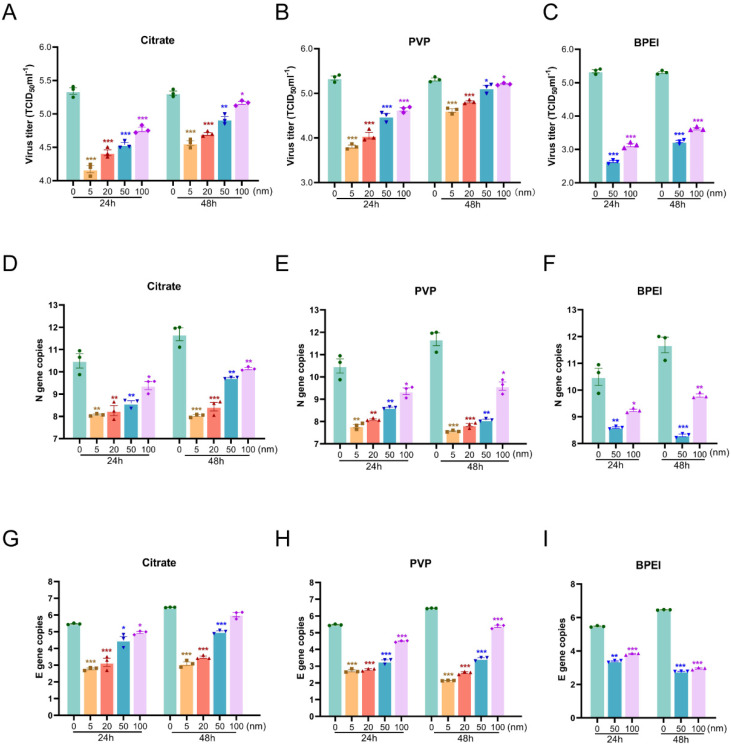
Effect of AgNPs on the replication of SARS-CoV-2 in Vero E6 cells. The cells were infected with SARS-CoV-2 (0.008 MOI). Vero E6 cells were treated with citrate, PVP, and BPEI coated AgNPs, with different particle sizes. The cell supernatant was collected at 24 and 48 h to detect the virus titer and the copy number of the virus genome. (**A**–**C**) Effects of different treatment times and particle sizes on SARS-CoV-2 virus titer of Citrate (**A**), PVP (**B**), and BPEI (**C**), respectively. (**D**,**E**) Effects of Citrate (**D**), PVP (**E**), and BPEI (**F**) with different particle sizes on the number of copies of the SARS-CoV-2 N gene, respectively. (**G**–**I**) effects of Citrate (**G**), PVP (**H**), and BPEI (**I**) with different particle sizes on the number of copies of the SARS-CoV-2 E gene, respectively. Three experiments were conducted (n = 3 group), * *p* <0.05, ** *p* < 0.01, *** *p* < 0.001.

**Figure 4 nanomaterials-12-00990-f004:**
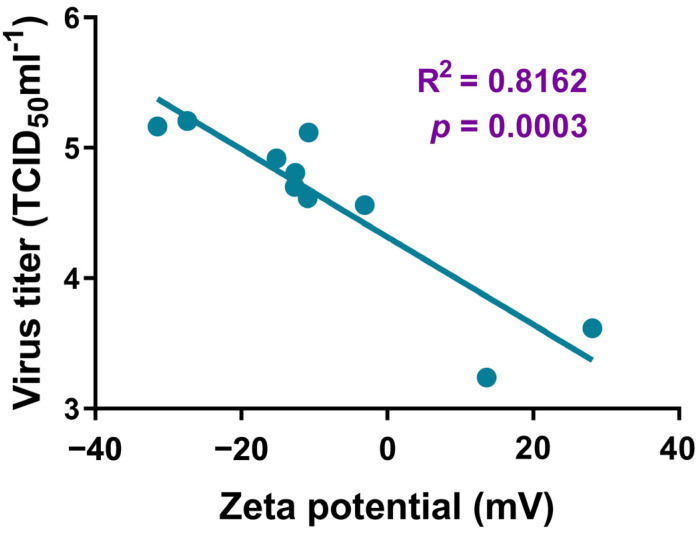
Correlation between the zeta potential of the modified material and virus titer.

**Figure 5 nanomaterials-12-00990-f005:**
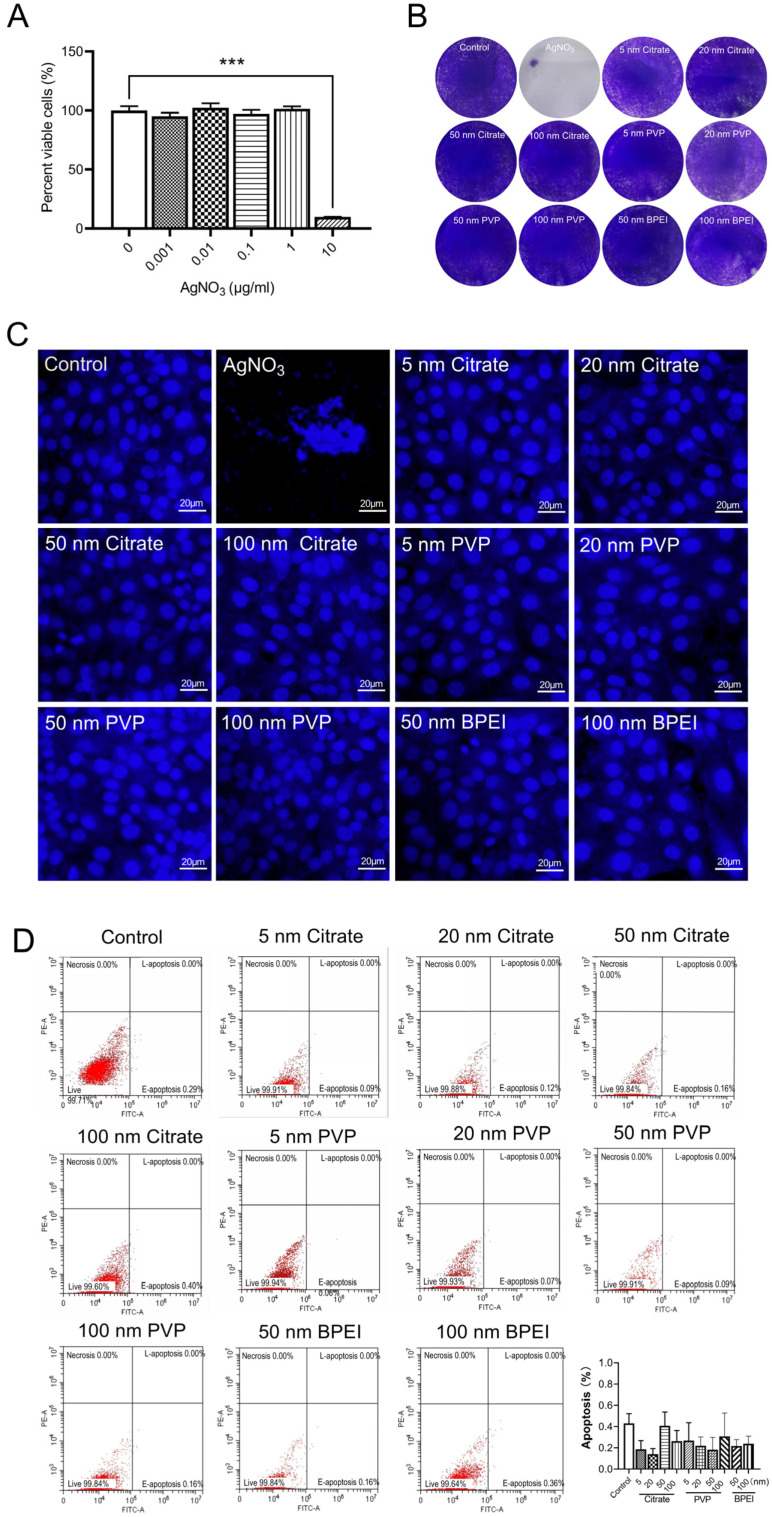
Evaluation of the cytotoxicity of AgNPs. (**A**) CCK-8 assays showing the effect of AgNO3 on the viability of Vero E6 cells (n = 8 groups); (**B**) Crystal violet staining of Vero E6 cells treated with AgNO3, citrate, PVP, and BPEI coated AgNPs, with different particle sizes; (**C**) Hoechst staining showing the effects of AgNO_3_, citrate, PVP, and BPEI coated AgNPs, with different particle sizes on the cell nucleus of Vero E6 cells, scale bar = 20 μm; (**D**) effects of citrate, PVP, and BPEI coated AgNPs on the apoptosis rate of Vero E6 cells, *** *p* < 0.001.

**Table 1 nanomaterials-12-00990-t001:** Characteristics of AgNPs.

AgNPs	Size by TEM (nm)	Hydrodynamic Size (nm)	Zeta Potential (mV)
PVP-5	6.0 ± 2.9	11.1 ± 1.4	−10.9 ± 4.3
PVP-20	18.9 ± 3.5	35.6 ± 6.6	−12.6 ± 1.4
PVP-50	48.8 ± 5.6	49.2 ± 1.0	−10.8 ± 1.2
PVP-100	96.2 ± 8.2	122.6 ± 1.0	−27.4 ± 0.6
Citrate-5	6.2 ± 1.8	7.8 ± 2.2	−3.1 ± 0.8
Citrate-20	22.0 ± 3.7	35.0 ± 14.6	−12.7 ± 1.0
Citrate-50	47.5 ± 5.8	62.2 ± 1.0	−15.2 ± 0.8
Citrate-100	114.1 ± 11.9	114.4 ± 0.5	−31.5 ± 0.6
BPEI-50	47.5 ± 5.3	71.4 ± 3.8	13.6 ± 1.0
BPEI-100	101.5 ± 10.9	114.3 ± 0.4	28.1 ± 0.4

Data are presented as mean ± SD.

## Data Availability

Data can be available upon request from the authors.

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
