# Peer review of "Antiviral Properties of Silver Nanoparticles against SARS-CoV-2: Effects of Surface Coating and Particle Size"

_nanomaterials, 2022, doi:10.3390/nano12060990_

Round 1
Reviewer 1 Report
Comments and Suggestions for Authors
I consider that the manuscript Antiviral properties of silver nanoparticles against SARS-CoV-2: Effects of surface coating and particle size should be consider for publication in Nanomaterials MDPI.
I have suggested some changes in the manuscript, which can be found in the attachment word file.
Questions:
- The Transmission Electron Microscopy of AgNPs has been included in the manuscript. Is it possible to include some reference or micrography to show the size and shape of the SARS-CoV-2 or more important the effect of nanoparticles?
- The effect of the particle size of AgNPs has been described in this work. What would happen with the shape of the NPs, would be the same with nanocubes, nanorods, nanostars, etc.?
Best Regards!!!

Author Response
Please see responses in attachment

Reviewer 2 Report
- The inroduction is not uptodate.
- Fig. 5 D please change the y-axis range in apoptosis curve.
- On my opinion, the application of your nanoparticles as external antimicrobial agent has no correlation with exams on infected cells.
Author Response
Please see responses in attachment

Reviewer 3 Report
- Give the particle and zeta potential image of the formulation.
- what is the rationale of experiments.
- how much is the IC 50 value.
- Add some research article related to the work in introduction section.
- Discusion part is poor Need to be revised and elaborated.
Author Response
Please see responses in attachment

Round 2
Reviewer 2 Report
On, my opinion there is no relevance between the experiments (infected cells) and the research aim (usage of AgNo3 as surface antimicrobal/nasal spray).
Reviewer 3 Report
accept
Author Response
Thanks for your favorable recommendation.